# Understanding Continuous-depth Networks through the Lens of Homogeneous Ricci Flows

## Abstract

The continuous-depth models pioneered by Neural ODEs have sparked a resurgence in exploring dynamic systems based on deep learning prototypes. The studies employed to investigate their theoretical properties mainly rely on Euclidean space, however, the geometric principle of general neural networks has been developed on the Riemannian manifold. Motivated by this open problem, we construct a formalized geometric theory of continuous-depth networks through the lens of homogeneous Ricci flows. From this perspective, the Riemannian metric tensor with coordinate representations learned by the continuous-depth network itself is the closed-form solution of homogeneous Ricci flows. With the presence of Ricci solitons, the Ricci curvature tensor on the underlying data manifold emerges for the first time. This implies that the continuous-depth network governs the Ricci curvature to drive the different kinds of data apart from each other, which is a novel observation between the Ricci curvature and data separation. Toy experiments confirm parts of the proposed theory, as well as provide intuitions and visualizations as to how the Ricci curvature tensor governed by continuous-depth networks evolves on the manifold to operate on data.

## 1 Introduction

In machine learning, neural networks have played a crucial role, including theoretical research and practical applications. The previous theoretical works about neural networks are built in Euclidean geometry, i.e., the input data is required to be immersed in $\mathbb{R}^n$ (Kidger & Lyons, 2020). However, from a geometric perspective, the Euclidean measurement may not always provide meaningful insights when the data points live on a lower-dimensional manifold (Kratsios & Bilokopytov, 2020), which is known as the manifold hypothesis. Indeed, the underlying structure of data is a non-Euclidean space in many cases of practical applications (Nickel & Kiela, 2017; Ganea et al., 2018). The main goal of manifold learning is to detect a low-dimensional representation that preserves meaningful information (Lin & Zha, 2008).

Recently, a new geometric framework has been proposed to view a neural network as a sequence of maps between smooth manifolds (Hauser & Ray, 2017; Shen, 2018). Specifically, while the input data is sampled from a data manifold, the neural network-driven coordinate transformation aims to measure the underlying data manifold by Euclidean distance (Hauser & Ray, 2017). This implies that the coordinate transformations learned by a neural network can represent the Riemannian metric tensor on the data manifold, thereby a neural network can be analyzed using Riemannian geometry. Based on this principle, Benfenati & Marta (2022) proposed that the pullbacks of the Riemannian metric through maps of a neural network are degenerate metrics such that establishing a singular Riemannian geometric framework. These degenerate metrics are able to endow each manifold of the sequence with the structure of pseudo-metric spaces, from which one can obtain a full-fledged metric space by means of metric identification. In practice, the geometric perspective also in turn guides the improvement of the network structure (Rousseau et al., 2020).

With the proposal of neural ordinary differential equations (Neural ODEs) (Chen et al., 2018), continuous deep learning (Zhang et al., 2014) ushered in a brand new beginning. Neural ODEs have been introduced as the limit of taking the number of layers of residual networks $\mathbf{h}_{i+1} = \mathbf{h}_i + f(\mathbf{h}_i, W_i)$ to infinity, resulting in continuous-depth networks $\frac{d\mathbf{h}(t)}{dt} = f(\mathbf{h}(t), t, W)$. Note that $\mathbf{h}$ is the hidden state of Neural ODEs, $f$ defines a first-order ODE, and $t \in [0, T]$. In addition, Neu-

ral ODEs also give new mathematical perspectives for neural network structure design (Greydanus et al., 2019; Bai et al., 2019; Cranmer et al., 2020). Furthermore, Neural ODEs have been demonstrated superior performance compared to traditional ODE inference techniques for non-linear dynamics $f$ (Dupont et al., 2019). Subsequently, SONODE extends Neural ODEs to second-order systems (Norcliffe et al., 2020). Some other continuous-depth networks have also been proposed, including Augmented NODE (Dupont et al., 2019), latent NODEs (Rubanova et al., 2019), and neural stochastic DEs (Volokhova et al., 2020; Li et al., 2020).

Although latent Neural ODEs are a powerful tool for processing and analyzing time series (Chen et al., 2018), the fundamental process of how successive layers continuously warp the coordinate representation of the data manifold to form the purpose that the classes in the data manifold are linearly separable is often overlooked. In this paper, we provide insight into continuous-depth networks from the perspective of time-evolving Riemannian geometry. Exploration in this uncharted field can help understand the geometric structure evolution of the data manifold, thereby optimizing the network structure and over-fitting problems. Our contributions are summarized as follows:

1. We propose a novel principle for continuous-depth networks through the lens of homogeneous Ricci flows, which will degenerate into the case of general neural networks (Hauser & Ray, 2017) after discretization. To the best of our knowledge, this is a first connection between continuous-depth models and Ricci flows, which mathematically provides a time-evolving geometric perspective for the interpretability of neural networks.

2. By introducing the self-similar solution (Ricci soliton) of the homogeneous Ricci flow, the Ricci curvature tensor, as a fundamental concept in the Riemannian geometry, can be learned by continuous-depth networks. In essence, the Ricci curvature may be the root in terms of geometry to drive the different kinds of data apart from each other.

3. It follows from Hauser & Ray (2017) that we provide a way of visualization and intuition to understand how the Ricci curvature tensor governed by continuous-depth models evolves on the underlying data manifold to untangle the input data.

## 2 PRELIMINARIES

### 2.1 RIEMANNIAN GEOMETRY

We will introduce the basic notions of manifolds and tangent spaces used in this paper. For more in-depth propositions, see Guggenheimer (2012); Petersen (2006). We then present some classic definitions and results from Riemannian geometry. We define a topological space as follows:

**Definition 1** *A **topological space** $\mathcal{T}$ is a Hausdorff space if there exists two neighborhoods $U_p, U_q$ on any two distinct points $p, q \in \mathcal{T}$ such that $U_p \cap U_q = \emptyset$. A topological space $\mathcal{T}$ is second countable if any open subset of $\mathcal{T}$ can be written as a union of elements of a countable collection $\mathcal{U} = \{U_i\}_{i=1}^{\infty}$.*

Then a smooth manifold can be defined on the basis of a topological space.

**Definition 2** *The $n$-dimensional **smooth manifold** $\mathcal{M}$ is a topological space, which satisfies second countable and Hausdorff. Then every point $p \in \mathcal{M}$ has a neighborhood $U_p$ that is homeomorphic to a subset of $\mathbb{R}^n$ through a map $\phi_p : U_p \to \mathbb{R}^n$. In particular, for $p, q \in \mathcal{M}$ if $U_p \cap U_q \neq \emptyset$, then $\phi_p \circ \phi_q^{-1}$ is a smooth **diffeomorphism**. And the pair $(U_p, \phi_p)$ is called a local **chart** and the collection of all the possible local charts at all points is called atlas.*

When a chart covers the whole manifold $\mathcal{M}$, one has a global chart (coordinate system) for the manifold. Given a manifold $\mathcal{M}$, a subset $\mathcal{N} \subset \mathcal{M}$ that is a manifold is called a submanifold of $\mathcal{M}$. Let us consider an $n$-dimensional manifold $\mathcal{M}$ with a coordinate system. Then the definition of tangent space is given as follows:

**Definition 3** *The **tangent space** of $\mathcal{M}$ at $p$ denoted with $T_p\mathcal{M}$ is the set of all tangent vectors at $p$, which is a vector space spanned by $n$ tangent vectors along the coordinate curves.*

Note that tangent vectors are not dependent on the choice of a chart. Intuitively, the tangent space can be regarded as a linearization of $\mathcal{M}$ in a neighborhood of $p$. Consequently, a fundamental concept:

Riemannian metric, can be defined on the basis of the tangent space so that Riemannian manifold is naturally well-defined.

**Definition 4** *A **Riemannian metric** $g$ over a smooth manifold $\mathcal{M}$ is a smooth family of inner products on the tangent spaces of $\mathcal{M}$, i.e., $g$ associates to every $p \in \mathcal{M}$ with an inner product $g_p : T_p\mathcal{M} \times T_p\mathcal{M} \to \mathbb{R}$.*

**Definition 5** *A **Riemannian manifold** $(\mathcal{M}, g)$ is a smooth manifold $\mathcal{M}$ with an inner product, defined by the symmetric and positive definite Riemannian metric $g = (g_{ab})$, varying smoothly on the tangent space $T_p\mathcal{M}$ of $\mathcal{M}$ where each point $p$ is on $\mathcal{M}$.*

In particular, all inner product spaces with the same dimension are isometric, therefore, all tangent spaces $T_p\mathcal{M}$ on a Riemannian manifold $(\mathcal{M}, g)$ are isometric to the $n$-dimensional Euclidean space $\mathbb{R}^n$ endowed with its canonical inner product.

We will use the Einstein summation notation in this paper, which is commonly used in differential manifolds. For example, $y^a z_a := \sum_a y^a z_a$ where the summation is over dummy index $a$. Then, some other basic concepts: curvatures, in a Riemannian manifold are yielded. The Christoffel symbol in terms of an ordinary derivative operator $\partial$ is defined as

$$\Gamma^k_{ij} = \frac{1}{2} g^{kl} (\partial_i g_{jl} + \partial_j g_{il} - \partial_l g_{ij})$$

Furthermore, the coordinate form of the Riemann curvature tensor (Rm) as a $(1, 3)$-tensor is given by

$$R^l_{ijk} = \partial_i \Gamma^l_{jk} - \partial_j \Gamma^l_{ik} + \Gamma^p_{jk} \Gamma^l_{ip} - \Gamma^p_{ik} \Gamma^l_{jp}$$

The Ricci curvature tensor (Rc), which is a $(0, 2)$-tensor, with coordinate expression is defined as a contraction of Rm on first and third indices

$$R_{ij} = R^p_{pij}$$

Sequentially, the scalar curvature (R) is defined as the trace of the Ricci curvature tensor with the Riemannian metric

$$R = g^{ij} R_{ij}$$

## 2.2 GEOMETRY IN DISCRETE-DEPTH NETWORKS

In this subsection, we introduce the Riemannian geometry of discrete-depth networks. A discrete-depth network composed of a sequence of transformations to a hidden state can be seen as the Euler discretization of a continuous transformation (Chen et al., 2018; Haber & Ruthotto, 2017).

A neural network can be regarded as a nonlinear function to transform the input $\boldsymbol{x}(0)$ into the output $\boldsymbol{x}(n)$ through $n$ layers. The corresponding transformation consists of two units: the first unit outputs the affine transformation via the parameter (weights and biases) $W_i$, and the second unit outputs the nonlinear transformation via an activation function, e.g., *ReLU*, *tanh*, etc. On the basis of Benfenati & Marta (2022), a geometric definition of discrete-depth networks is viewed as a sequence of smooth maps $\phi_j$, where $j \in \{1, 2, \cdots, n\}$, between smooth manifolds $\mathcal{M}_i$ of the form:

$$\mathcal{M}_0 \xrightarrow{\phi_1} \mathcal{M}_1 \xrightarrow{\phi_2} \cdots \xrightarrow{\phi_{n-1}} \mathcal{M}_{n-1} \xrightarrow{\phi_n} \mathcal{M}_n, \tag{1}$$

where $\mathcal{M}_0$ is the input manifold and $\mathcal{M}_n$ is the output manifold. As per the findings presented in (Hauser & Ray, 2017), the data points supplied to a neural network have the potential to be depicted in Cartesian coordinates, which are sampled from the underlying input manifold. And the parameter $W_i$ serves as a coordinate chart on the parameter manifold (Amari & Nagaoka, 2000). The coordinate representation of the data manifold is learned by neural networks to linearly separate the data such that the output coordinates are required to be a flattened representation of the data manifold (Bengio et al., 2013), which implies the output coordinate is measured by Euclidean distance when the output as a classifier has been well-trained by the neural network. Therefore, the Euclidean metric $g^E$ is implicit in the output coordinates:

$$g(\boldsymbol{x}(n))_{a_n, b_n} = g^E_{a_n, b_n}. \tag{2}$$

**Example 1** *The Euclidean metrics of $\mathbb{R}^2$ in Cartesian coordinates $(x, y)$ and polar coordinates $(r, \theta)$ are represented by the matrix respectively*

$$g^C = \begin{pmatrix} 1 & 0 \\ 0 & 1 \end{pmatrix}, \quad g^P = \begin{pmatrix} 1 & 0 \\ 0 & r^2 \end{pmatrix}.$$

Consider two adjacent smooth manifolds $\mathcal{M}_i$, $\mathcal{M}_{i+1}$ equipped with the Riemannian metrics $g(i)$, $g(i+1)$ and let $\phi : \mathcal{M}_i \to \mathcal{M}_{i+1}$ be a smooth map. Then these two Riemannian metrics can be connected by the pullback $\phi^*$ through $\phi$, i.e., $g(i) = \phi^* g(i+1)$. Given two coordinate systems $\boldsymbol{x}(i)$ and $\boldsymbol{x}(i+1)$ of $\mathcal{M}_i$ and $\mathcal{M}_{i+1}$ respectively, the metric in $i$-th coordinate system can be expressed by the metric in $(i+1)$-th coordinate system:

$$
\begin{aligned}
g(\boldsymbol{x}(i))_{a_i,b_i} &= \sum_{a_{i+1},b_{i+1}}^{\dim(\mathcal{M}_{i+1})} \left(\frac{\partial \phi}{\partial \boldsymbol{x}(i)}\right)_{a_i}^{a_{i+1}} g(\boldsymbol{x}(i+1))_{a_{i+1},b_{i+1}} \left(\frac{\partial \phi}{\partial \boldsymbol{x}(i)}\right)_{b_i}^{b_{i+1}} \\
&= (\boldsymbol{J}_i)_{a_i}^{a_{i+1}} (\boldsymbol{J}_i)_{b_i}^{b_{i+1}} g(\boldsymbol{x}(i+1))_{a_{i+1},b_{i+1}},
\end{aligned}
\tag{3}
$$

where $\phi = \boldsymbol{x}(i+1) = \boldsymbol{x}(i) + f(\boldsymbol{x}(i), W_i)$ in residual networks and $\boldsymbol{J}_i$ is the Jacobian matrix in local coordinates, i.e., $(\boldsymbol{J}_i)_{a_i}^{a_{i+1}} = \left(\frac{\partial \phi}{\partial \boldsymbol{x}(i)}\right)_{a_i}^{a_{i+1}}$. Note that we use the Einstein summation to simplify the summation nation. Furthermore, the metric in $i$-th coordinate system can also be directly expressed by the Euclidean metric in the output coordinate system based on Eq.(2) and Eq.(3):

$$
g(\boldsymbol{x}(i))_{a_i,b_i} = \prod_i^{n-1} \left[ \left(\frac{\partial \phi}{\partial \boldsymbol{x}(i)}\right)_{a_i}^{a_{i+1}} \left(\frac{\partial \phi}{\partial \boldsymbol{x}(i)}\right)_{b_i}^{b_{i+1}} \right] g_{a_n,b_n}^E = \prod_i^{n-1} \left[ (\boldsymbol{J}_i)_{a_i}^{a_{i+1}} (\boldsymbol{J}_i)_{b_i}^{b_{i+1}} \right] g_{a_n,b_n}^E.
\tag{4}
$$

In this case, the Riemannian metric tensor with the coordinate representation tends to be Euclidean from the input manifold to the output manifold.

## 3 TIME-EVOLVING GEOMETRY IN CONTINUOUS-DEPTH NETWORKS

### 3.1 HOMOGENEOUS RICCI FLOW AND RICCI SOLITON

Previous works (Hauser & Ray, 2017; Benfenati & Marta, 2022) build the geometry around discrete-depth networks. When considering a time-dependent continuous-depth network, we naturally need to rebuild the geometric principle in terms of flow. An intuitive idea is to introduce the Ricci flow since it is a partial differential equation (PDE) for the evolution of manifolds, involving the alteration of the manifold's metric to reveal changes in its geometric and topological properties over time. The Ricci flow was first published by Hamilton Hamilton et al. (1982), whose purpose is to prove Thurston's Geometrization Conjecture and Poincaré Conjecture by evolving the metric to make the manifold become "round". Naturally, the metric will evolve towards certain fundamental geometric structures, and the connected sum decomposition by sphere and tori will somehow emerge Sheridan & Rubinstein (2006), which is performed "surgery" on the manifold Perelman (2003). Given a Riemannian manifold $\mathcal{M}$ with a time-dependent metric $g(t)$, the Ricci flow yields

$$\frac{\partial}{\partial t} g(t) = -2 \operatorname{Rc}[g(t)], \tag{5}$$

where $\operatorname{Rc}$ denotes the Ricci curvature tensor and $g(0) = g_0$ is the initial metric.

Given a transitive Lie group $G \subset \mathrm{I}(\mathcal{M}, g_0)$ and a isotropy subgroup $K \subset G$ at some point $p \in \mathcal{M}$, a homogeneous space $G/K$ is represented as

$$(\mathcal{M}, g(t)) = (G/K, g_{\langle\cdot,\cdot\rangle}(t)), \quad \text{with the same reductive decomposition} \quad \mathfrak{g} = \mathfrak{k} \oplus \mathfrak{p} \tag{6}$$

where $\mathfrak{g}$ and $\mathfrak{k}$ are the Lie algebras of $G$ and $K$, respectively. And $\mathfrak{p}$ can be naturally identified with the tangent space $\mathfrak{p} = T_p G/K$. When the homogeneous space is presented, we can denote $g_{\langle\cdot,\cdot\rangle}(t)$ as the $G$-invariant metric on $G/K$ Duistermaat & Kolk (2012). This $G$-invariant metric is completely determined by its value at the origin $g_{\langle\cdot,\cdot\rangle}(t)|_p$, which is an $\operatorname{Ad}(K)$-invariant inner product on $\mathfrak{p}$ defined by $g$.

**Homogeneous Ricci flow**    For the family $g_{\langle\cdot,\cdot\rangle}(t)$, the Ricci flow in homogeneous spaces reduces to the ODE Lauret (2013)

$$\frac{d}{dt}g_{\langle\cdot,\cdot\rangle}(t) = -2\,\mathrm{Rc}\left[g_{\langle\cdot,\cdot\rangle}(t)\right].\tag{7}$$

And the solution $g_{\langle\cdot,\cdot\rangle}(t)$ to homogeneous Ricci flow still stays $\mathrm{Ad}(K)$-invariant for all $t$ and guarantees the short time existence and uniqueness.

**Definition 6** *Given two homogeneous spaces $G_0/K_0$ and $G_t/K_t$, a differentiable map $\phi$ : $G_0/K_0 \to G_t/K_t$ is called a **diffeomorphism** if it is a bijection and its inverse $\phi^{-1} : G_t/K_t \to G_0/K_0$ is differentiable as well.*

**Remark 1** *Two manifolds $G_0/K_0$ and $G_t/K_t$ are diffeomorphic (usually denoted $G_0/K_0 \simeq G_t/K_t$) if there is a diffeomorphism $\phi$ from $G_0/K_0$ to $G_t/K_t$.*

**Remark 2** *Considering the properties of diffeomorphisms, the activation function is required to be diffeomorphic. For example, tanh and sigmoid are satisfied, but ReLU is not satisfied.*

All the manifolds including the input $G_0/T_0$ and output $G_T/K_T$ are homogeneous, then we have the time-dependent family of diffeomorphisms (with $\phi(0) = \mathrm{id}$) $\phi(t) : G_0/K_0 \to G_t/K_t$ between homogeneous spaces. In this case, $\phi(t)$ as the equivariant diffeomorphism is determined by a Lie group isomorphism between $G_0$ and $G_t$.

**Homogeneous Ricci soliton**    However, the evolution of Ricci flow is very complex and often develops singularities, which makes it difficult to be solved. Hence, we turn our attention to Ricci solitons that are special solutions of the Ricci flow. More precisely, a Ricci soliton $(\mathcal{M}, g)$ yields a self-similar solution to the Ricci flow equation, that is, only by scaling and pullback by diffeomorphisms (Lafuente & Lauret, 2014).

**Definition 7** *A homogeneous space $(G/K, g_{\langle\cdot,\cdot\rangle})$ is called a **homogeneous Ricci soliton** if, and only if, there exists a smooth vector field $V$ such that*

$$\mathrm{Rc}(g_{\langle\cdot,\cdot\rangle}) = \lambda g_{\langle\cdot,\cdot\rangle} - \frac{1}{2}\mathcal{L}_V g_{\langle\cdot,\cdot\rangle} = \lambda g_{\langle\cdot,\cdot\rangle} - \frac{1}{2}\lim_{\delta t \to 0}\frac{(\phi_V^{\delta t})^* g_{\phi_V^{\delta t}\langle\cdot,\cdot\rangle} - g_{\langle\cdot,\cdot\rangle}}{\delta t},\tag{8}$$

*for some constant $\lambda \in \mathbb{R}$. Here $\mathcal{L}$ represents the Lie derivative.*

It is worth mentioning that, up to diffeomorphism and depending on the sign of $\lambda$, a homogeneous Ricci soliton homothetically shrinks ($\lambda > 0$), remains steady ($\lambda = 0$) or expands ($\lambda < 0$) under homogeneous Ricci flow. Later, we will focus on the case that ricci solitons are steady.

## 3.2    THE SOLUTION OF HOMOGENEOUS RICCI FLOWS

Inspired by previous work (Chen et al., 2021), we consider that both Ricci flow and continuous-depth networks serve the same purpose, i.e., to continuously evolve the input manifold such that the output manifold has specific geometric properties. Naturally, we can establish the geometry principle for continuous-depth networks on the basis of homogeneous Ricci flow and Ricci soliton. Of course, we can proceed with the discretization to check whether the geometric principle degenerates into a discrete case (Hauser & Ray, 2017), and then evaluate the rationality of the proposed theory.

**Ricci curvature tensor using homogeneous Ricci soliton**    We consider that a homogeneous Ricci soliton is steady. Then the Ricci curvature tensor yields $-2\,\mathrm{Rc}(g_{\langle\cdot,\cdot\rangle}) = \mathcal{L}_V g_{\langle\cdot,\cdot\rangle} = \lim_{\delta t \to 0}\frac{1}{\delta t}\left((\phi_V^{\delta t})^* g_{\phi_V^{\delta t}\langle\cdot,\cdot\rangle} - g_{\langle\cdot,\cdot\rangle}\right)$. The Lie derivative is the speed with which the Riemannian metric tensor changes under the space deformation caused by the flow. Formally, given a differentiable (time-independent) vector field $V$ on a homogeneous space $G/K$, let $\phi_V^{\delta t} : G/K \to G/K$ be the corresponding local flow. Since $\phi_V^{\delta t}$ is a local diffeomorphism for each $\delta t$, it gives rise to a pullback of metrics. Since the diffeomorphism is given by the neural network, we can define the Lie derivative using Eq.(3). Consequently, we formally define the Ricci curvature tensor at time $t$ of

continuous-depth networks:

$$
\begin{aligned}
&\mathrm{Rc}(\boldsymbol{x}(t))_{a_t,b_t} \\
&= \lim_{\delta t \to 0} -\frac{1}{2\delta t} \left( g_{\langle \cdot,\cdot \rangle}(\boldsymbol{x}(t - \delta t))_{a_{t-\delta t},b_{t-\delta t}} - g_{\langle \cdot,\cdot \rangle}(\boldsymbol{x}(t))_{a_t,b_t} \right) \\
&= \lim_{\delta t \to 0} -\frac{1}{2\delta t} \left( \left( \frac{\partial \phi}{\partial \boldsymbol{x}(t-\delta t)} \right)^{a_t}_{a_{t-\delta t}} \left( \frac{\partial \phi}{\partial \boldsymbol{x}(t-\delta t)} \right)^{b_t}_{b_{t-\delta t}} g_{\langle \cdot,\cdot \rangle}(\boldsymbol{x}(t))_{a_t,b_t} - g_{\langle \cdot,\cdot \rangle}(\boldsymbol{x}(t))_{a_t,b_t} \right) \\
&= \lim_{\delta t \to 0} -\frac{1}{2\delta t} \left( (\boldsymbol{J}_{t-\delta t})^{a_t}_{a_{t-\delta t}} (\boldsymbol{J}_{t-\delta t})^{b_t}_{b_{t-\delta t}} g_{\langle \cdot,\cdot \rangle}(\boldsymbol{x}(t))_{a_t,b_t} - g_{\langle \cdot,\cdot \rangle}(\boldsymbol{x}(t))_{a_t,b_t} \right) \\
&= \lim_{\delta t \to 0} -\frac{1}{2\delta t} \left( (\boldsymbol{J}_{t-\delta t})^{a_t}_{a_{t-\delta t}} (\boldsymbol{J}_{t-\delta t})^{b_t}_{b_{t-\delta t}} - \boldsymbol{I} \right) g_{\langle \cdot,\cdot \rangle}(\boldsymbol{x}(t))_{a_t,b_t},
\end{aligned}
\tag{9}
$$

where $\phi = \boldsymbol{x}(t)$ and $\boldsymbol{I}$ is a identity matrix that can be written as $\boldsymbol{I} = (\boldsymbol{J}_t)^{a_t}_{a_t}(\boldsymbol{J}_t)^{b_t}_{b_t}$. Let the depth of a neural network to be $T$, then a homogeneous Ricci flow is to evolve the manifold on the time $t \in [0, T]$. Substituting the above formula into Eq.(7), the homogeneous Ricci flow in a homogeneous Ricci soliton yields

$$
\frac{d}{dt} g_{\langle \cdot,\cdot \rangle}(t) = \frac{1}{\delta t} \left( \boldsymbol{J}_{t-\delta t} \boldsymbol{J}_{t-\delta t} - \boldsymbol{I} \right) g_{\langle \cdot,\cdot \rangle}(t).
\tag{10}
$$

**Riemannian metric using homogeneous Ricci flow**    Mathematically, we can solve for the Riemannian metric $g_{\langle \cdot,\cdot \rangle}(t)$ by integrating the above formula from $t$ to $T$. Note that the metric $g_{\langle \cdot,\cdot \rangle}(T)$ of the output manifold is set in advance, e.g., the Euclidean metric on the basis of Eq.(2). We have

$$
\begin{aligned}
&g_{\langle \cdot,\cdot \rangle}(\boldsymbol{x}(t))_{a_t,b_t} \\
&= g_{\langle \cdot,\cdot \rangle}(\boldsymbol{x}(T))_{a_T,b_T} \exp\left( \int_T^t \frac{1}{\delta t} \left( \left( \frac{\partial \phi}{\partial \boldsymbol{x}(t-\delta t)} \right)^{a_t}_{a_{t-\delta t}} \left( \frac{\partial \phi}{\partial \boldsymbol{x}(t-\delta t)} \right)^{b_t}_{b_{t-\delta t}} - \boldsymbol{I} \right) dt \right) \\
&= g_{\langle \cdot,\cdot \rangle}(\boldsymbol{x}(T))_{a_T,b_T} \exp\left( \int_T^t \frac{1}{\delta t} \left( (\boldsymbol{J}_{t-\delta t})^{a_t}_{a_{t-\delta t}} (\boldsymbol{J}_{t-\delta t})^{b_t}_{b_{t-\delta t}} - \boldsymbol{I} \right) dt \right) \\
&= \exp\left( \mathrm{ODESolve}\left( \log(g_{\langle \cdot,\cdot \rangle}(\boldsymbol{x}(T))_{a_T,b_T}), \left( (\boldsymbol{J}_{t-\delta t})^{a_t}_{a_{t-\delta t}} (\boldsymbol{J}_{t-\delta t})^{b_t}_{b_{t-\delta t}} - \boldsymbol{I} \right) / \delta t, t, T, W \right) \right),
\end{aligned}
\tag{11}
$$

where this numerical integral can be solved by a Neural ODE solver (Chen et al., 2018).

Now, we introduce Type I product integral corresponding to Volterra's original definition Dollard & Friedman (1979); Slavík (2007)

$$
\prod_a^b (\boldsymbol{I} + \boldsymbol{S}(t)\delta t) = \exp\left( \int_a^b \boldsymbol{S}(t) dt \right),
\tag{12}
$$

which can be used to check the rationality of the proposed theory. In particular, $\delta t = 1$ holds in discrete-depth networks. There is no doubt that after the discretization, Eq.(11) will degenerate into Eq.(4), which proves that our geometric principle is a generalization of previous works (Hauser & Ray, 2017; Benfenati & Marta, 2022).

## 4   THE RICCI CURVATURE TENSOR GOVERNED BY NEURAL NETWORKS

We introduced the Riemannian metric tensor by solving the homogeneous Ricci flow. And the most important thing is that we introduced another concept in Riemannian geometry: Ricci curvature tensor. The corresponding pseudo-code is described in Algorithm 1. In this section, we delve into an exploration of neural network behavior through the lens of the homogeneous Ricci flow. The central idea is to scrutinize the connection between neural networks and the evolution of manifolds, shedding light on the theoretical underpinnings that untangle the data by the learning.

As shown in Figure 1, we express $(G_0/K_0, g_{\langle \cdot,\cdot \rangle}(0))$ and $(G_T/K_T, g_{\langle \cdot,\cdot \rangle}(T))$ as the input and output manifolds, which we interpret as the initial and final states of a process, respectively. Meanwhile, we view the other states as intermediate stages in the evolution of these manifolds. This portrayal

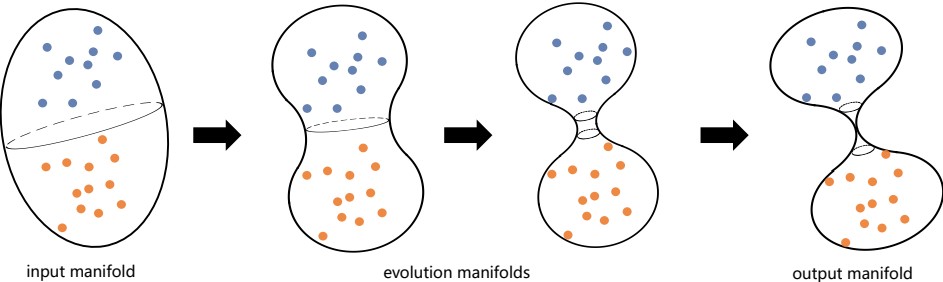

input manifold  evolution manifolds  output manifold

Figure 1: The diagram illustrates that the manifold will gradually "pinch off" in terms of the evolution of the Ricci flow, where the black line represents this 2-manifold. From the input manifold to the output manifold, the curvature gradually swells near the hyperplanes, and with that comes the manifold pinching, which tends to decompose into two sub-manifolds.

allows us to envision how the positions of the blue and orange points transform as the manifold evolves. The visual aid provided by the black line, representing a 2-manifold, helps convey this concept more vividly. In the transition from the input manifold to the output manifold, an intriguing phenomenon emerges. The curvature of the manifold exhibits a gradual swelling in proximity to the hyperplanes. This curvature evolution initiates a process known as "manifold pinching" (Sheridan & Rubinstein, 2006), wherein the manifold becomes increasingly constricted and tends to split into two distinct sub-manifolds. It's within this framework that we observe the neural network's capacity to segregate data. In essence, the Ricci curvature serves as the foundational geometrical attribute that enables neural networks to carry out the separation of data categories. This deep connection between the Ricci curvature and neural network behavior opens avenues for further exploration at the intersection of geometry and machine learning.

---

**Algorithm 1** Pseudo-code for understanding continuous-depth networks.

---

**Require:** parameters $W$, start time $0$, stop time $T$, intermediate state $\boldsymbol{x}(t)$.
**Ensure:** The continuous-depth network has been well trained.
  1: Set the metric $g_{\langle \cdot, \cdot \rangle}(T)$ of the output manifold;
  2: Compute the Jacobian, i.e., $\boldsymbol{J}_{t-\delta t} = \left( \frac{\partial \boldsymbol{x}(t)}{\partial \boldsymbol{x}(t-\delta t)} \right)^{a_t}_{a_{t-\delta t}}$ and $\boldsymbol{J}_{t-\delta t} = \left( \frac{\partial \boldsymbol{x}(t)}{\partial \boldsymbol{x}(t-\delta t)} \right)^{b_t}_{b_{t-\delta t}}$;
  3: Compute the Riemannian metric tensor $g_{\langle \cdot, \cdot \rangle}(t) = g_{\langle \cdot, \cdot \rangle}(T) \exp \left( \int_T^t \frac{1}{\delta t} \left( \boldsymbol{J}_{t-\delta t} \boldsymbol{J}_{t-\delta t} - \boldsymbol{I} \right) dt \right)$;
  4: Compute the Ricci curvature tensor $\mathrm{Rc}(t) = -\frac{1}{2\delta t} \left( \boldsymbol{J}_{t-\delta t} \boldsymbol{J}_{t-\delta t} - \boldsymbol{I} \right) g_{\langle \cdot, \cdot \rangle}(t)$;

---

## 5 NUMERICAL EXPERIMENTS

In this section, it follows from Hauser & Ray (2017) that we yield similar 2D visualizations to show two geometric quantities: Riemannian metric and Ricci curvature tensor corresponding to the untangling process of a continuous-depth network. Note that this paper only provides theoretical insights and does not involve improvements to the network structure. **Since the experiments are only for the visualization and interpretability of the proposed theory, they are conducted on toy data (spiral and two circles) rather than real data**, where the two circles are from sklearn library and the spiral is achieved from `https://gist.github.com/45deg`. All the experiments implemented in Python are conducted with PyTorch.

**2-dimensional spiral and circles datasets.**   We generated a dataset of 1000 2-dimensional spirals and circles. For two types of spirals, half are clockwise while the other half are counter-clockwise. For two types of circles, data points within one circle belong to one category, while data points within the other circle belong to a different category. To introduce diversity and simulate real-world imperfections, we introduced Gaussian noise with a standard deviation of 0.02.

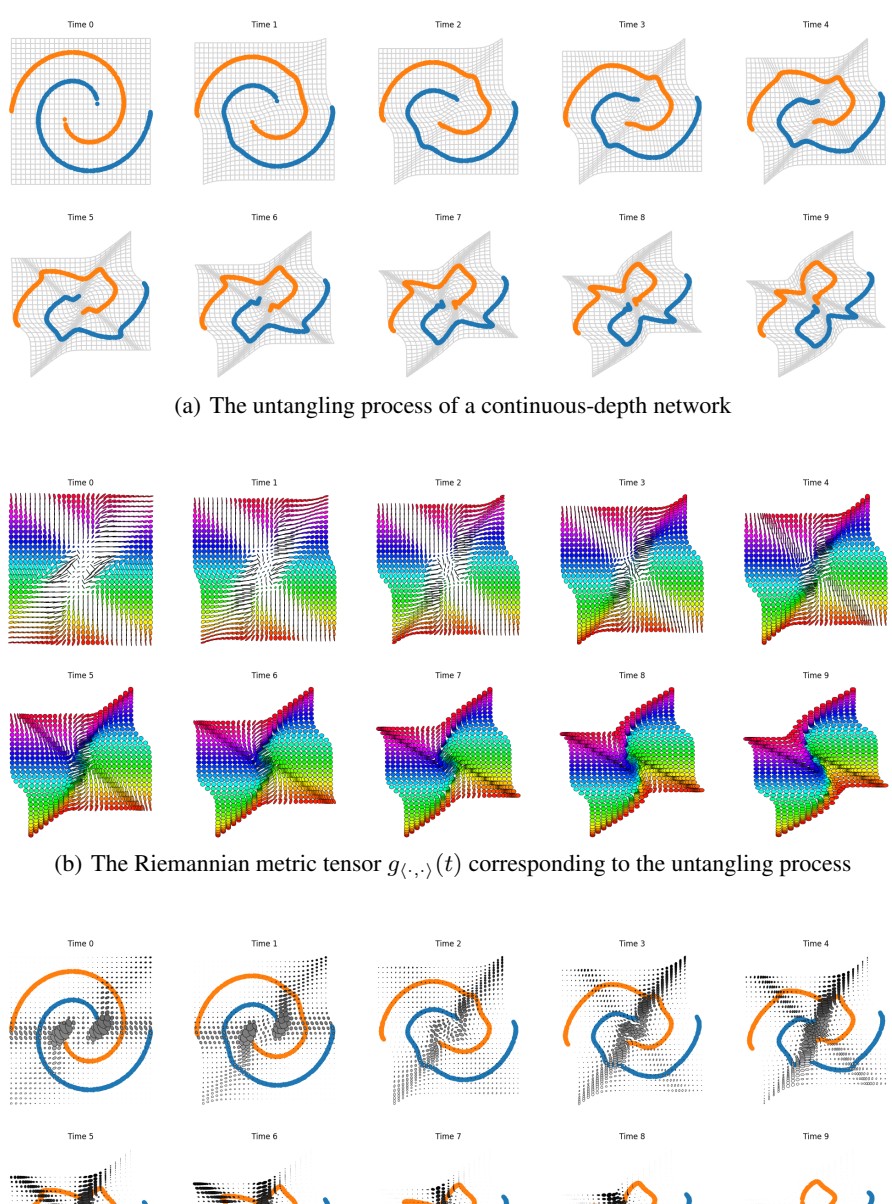

(a) The untangling process of a continuous-depth network

(b) The Riemannian metric tensor $g_{\langle\cdot,\cdot\rangle}(t)$ corresponding to the untangling process

(c) The Ricci curvature tensor $\mathrm{Rc}(t)$ corresponding to the untangling process

Figure 2: The evolution of the homogeneous Ricci flow is governed by a continuous-depth network to untangle the spiral data, where the gray ellipses represent the Ricci curvature tensor that came from the right hand side of the Ricci flow and the color ellipses represent the Riemannian metric tensor that came from the left side of the Ricci flow.

**Experiment details.** In the experiments, in order to ensure that each layer has 2D visualization, we choose the linear layer of a neural network as nn.Linear(2,2) and the output layer as nn.Linear(2,1) in Pytorch. We train with 200 epochs and adopt the Adam optimizer with a weight decay of 0.05 where the learning rate is 0.01. Note that the numerical approximation of Algorithm 1 can be achieved by the Neural ODE solver at `https://github.com/rtqichen/torchdiffeq`.

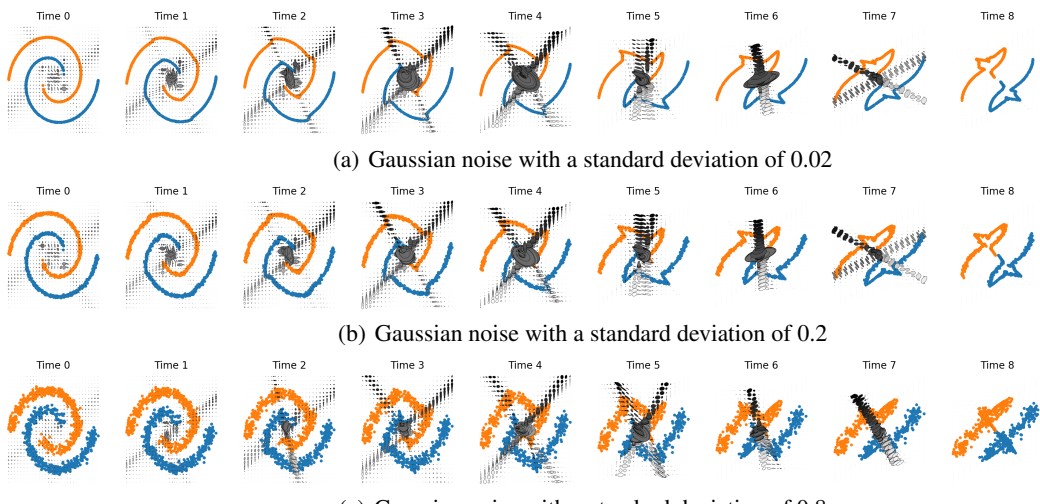

(a) Gaussian noise with a standard deviation of 0.02

(b) Gaussian noise with a standard deviation of 0.2

(c) Gaussian noise with a standard deviation of 0.8

Figure 3: Comparison of the Ricci curvature tensor on the spiral data with different noise. As the noise increases to a certain extent, a continuous-depth network can still govern the Ricci curvature to evolve the data manifold such that untangling the spiral data.

**Untangling process, Ricci curvature, and Riemannian metric.** In this experiment, we visualize the untangling process, and the corresponding Riemannian metric and Ricci curvature tensor by uniformly sampling several time nodes, where the size and direction of the ellipse represent the size and angle of the tensor. Note that the untangling process in Figure 2(a) and geometric quantities in Figure 2(b),2(c) are in one-to-one correspondence at the same time node.

In Figure 2, it seems that the homogeneous Ricci flow evolves on a manifold into the analog of hyperplanes to push the orange and blue points away from each other. Just like the characteristics described by Ricci flow, the manifold will "pinch off" near the hyperplanes and gradually decompose into two sub-manifolds to complete the untangling. This implies that the swelling of the curvature is gradually concentrated near hyperplanes. In the stop time (Time 9) as shown in Figure 2(b), distances are measured by the Euclidean metric and so the ellipses (metric) are "round". Especially in Figure 2(c), the Ricci curvature tensor tends to become particularly large where the two types of points are entangled closely. And it tends to become small where the two types of points are relatively far apart. Until the stop time (Time 9), the ellipses (curvature) are "zero". Intuitively, we consider that the Ricci curvature may be the root in terms of geometry that continuous-depth networks are able to drive the different kinds of data apart from each other.

**Effect of noise on the Ricci curvature.** In this experiment, we present the Ricci curvature and the corresponding untangling behavior of a continuous-depth network under different noise and coordinate representations. In this case, a continuous-depth network with several hidden layers has been well-trained such that the data points can be untangled with 0% error rate. In Figure 3, as the noise increases to a certain extent, a continuous-depth network can still govern the Ricci curvature to evolve the data manifold such that untangling the spiral data.

## 6 CONCLUSION

This paper presents a novel time-dependent geometric perspective on the untangling behavior of continuous-depth networks through the lens of homogeneous Ricci flows. By introducing the self-similar solution (Ricci soliton) of the homogeneous Ricci flow, the Ricci curvature tensor, as a fundamental concept on the Riemannian manifold, can be found by the coordinate representations learned by continuous-depth networks. In essence, we consider that the Ricci curvature may be the root in terms of geometry that continuous-depth networks are able to drive the different kinds of data apart from each other. In our future work, we intend to explore practical computer vision tasks, leveraging the theoretical insights acquired from Ricci flow in deep neural networks.

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

## A    EFFECT OF COORDINATE CHOICE ON THE RICCI CURVATURE

In Figure 4, a continuous-depth network in Cartesian coordinates takes a long time to evolve the manifold to untangle the points of two classes on the circles. In contrast, a continuous-depth network in polar coordinates can easily untangle the points of two classes on the circles. And the Ricci curvature on the data manifold is gradually swelling, especially in the middle of the two types of data, which further pushes the two types of data away.

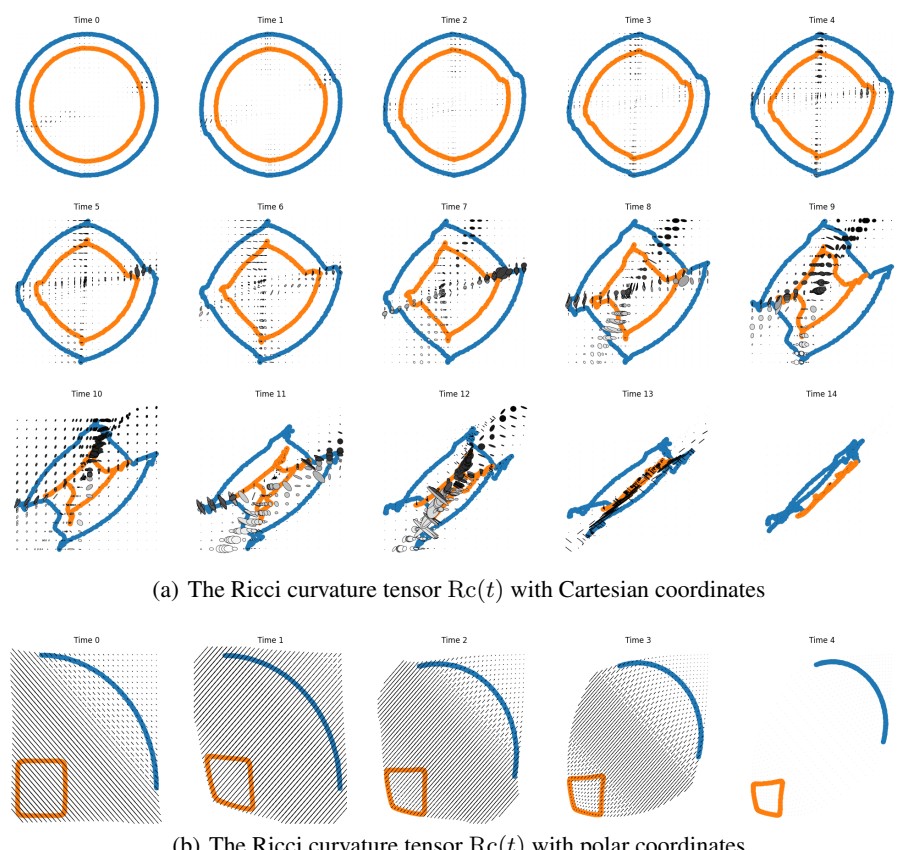

(a) The Ricci curvature tensor $\mathrm{Rc}(t)$ with Cartesian coordinates

(b) The Ricci curvature tensor $\mathrm{Rc}(t)$ with polar coordinates

Figure 4: Comparison of the Ricci curvature tensor with different coordinate representations on the two circles data. In Cartesian coordinates, the Ricci curvature governed by a continuous-depth network seems to be difficult to untangle the two circles data. In contrast, the two circles data can be easily untangled in polar coordinates.

