# OpenReview forum: "Understanding Continuous-depth Networks through the Lens of Homogeneous Ricci Flows"
_ICLR.cc/2024/Conference — Submitted to ICLR 2024_

### Official Review · Reviewer_8HH4 · 2023-10-30

**Soundness:** 1 poor
**Presentation:** 1 poor
**Contribution:** 2 fair
**Rating:** 3
**Confidence:** 3

**Summary:**

This papper invesitages continuous depth networks with the techniques of Ricci flow. This submission provides sufficiently amount of experiments. The homogeneous Ricci flows provides a explanation on continuous depth networks, which bridges for the first time the neural networks and Ricci flow. Furthermore, it is shown that Ricci soliton and Ricci curvature tensor can be learnt by continuous depth networks.

**Strengths:**

This submission has provdied good illustrations of the evolving process, and a new perspective for understanding the interpretability of neural networks intuitively, which makes this paper novel.

**Weaknesses:**

Despite the claims of contribution, it is not clear that what is the main technical/theoretical results this paper has provided. This is mainly caused by the presentation and organization. It seems that no theoretical guarantee has been shown in any place in the main article.

It could be an interesting direction that Ricci flow and discrete depth networks come to a common ground, and I believe this is not well understood by the machine learning community. However, the layout of section 3 is far from being able to attract general audience of this venue.

**Questions:**

What is the main theoretical claim that the experiments are verifying for?

---

### Official Review · Reviewer_6G4r · 2023-10-31

**Soundness:** 2 fair
**Presentation:** 1 poor
**Contribution:** 3 good
**Rating:** 3
**Confidence:** 2

**Summary:**

This work presents a novel geometric perspective of continuous-depth neural networks by using established tools from differential geometry such as the homogeneous Ricci flow to show how neural networks shape the underlying Ricci curvature of the representation space. The authors verify their theoretical contributions by visualising the evolution of the Ricci curvature and how it indeed leads to a separation of the data.

**Strengths:**

1. Studying neural networks from a more geometric perspective is a promising avenue that could lead to novel insights into the inner workings of neural networks. Leveraging the powerful tools from differential geometry such as Ricci flows and Ricci solitons is a helpful contribution that could foster more such works.
2. The visualisations of the underlying tensors are very interesting and display that the theory indeed seems to be capturing the essence of representation learning.

**Weaknesses:**

The paper is really hard to follow and the authors don’t do a great job at explaining the (admittedly complex) concepts needed in this work. What made it however even tougher for me to follow is that it is very difficult to understand when and where exactly the structure of the neural network is actually used in the theory. The authors start with a rather abstract introduction to Ricci flow, Ricci solitons etc, which is great to have, but then the connection to continuous-depth networks happens very suddenly and in an unclear manner. What exactly is the structure of the neural network determining in the equations? Or differently put, which quantities previously kept abstract (e.g. the manifold, the metric tensor at time $t$, the Ricci curvature etc) is now determined by assuming a neural network structure? I believe it is the diffeomorphisms?

Also, why are we working with the homogeneous Ricci flow instead of the standard Ricci flow? The authors also use a lot of concepts without introducing them, what is ad(K)-invariant for instance? Such things might be obvious to researchers closely working in this field, but even for people interested in theoretical ML research, this paper is really tough to read. There are lots of statements like “There is no doubt that after the discretisation, Eq 11 will degenerate into Eq 4” that are not obvious to me at least.
\
\
I think the ideas in this work are interesting, but the work in its current shape really does not explain them well, making it very difficult for me to assess this work positively. I’m happy to re-consider my score if the authors can clarify and potentially incorporate my feedback.

**Questions:**

1. On page 6, when you define the Ricci curvature as the Lie derivative (equation 9), in the third line of the derivation, where did the diffeomorphisms $\phi_{V}^{\delta t}$ and its pullback $(\phi_{V}^{\delta t})^{*}$ go? How are they defined in case of a neural network? I guess the diffeomorphisms is simply the forward pass from time $0$ to time $\delta t$?
2. For equation (10) right hand side, where did the limit $\lim_{\delta t \xrightarrow{} 0}$ go? Similarly in equation (12), the left-hand side seems to depend on $\delta t$ while the right-hand side does not. There is also no dependence on $a$ in the product.
3. Where in the theory do you explicitly use the fact that the output space is Euclidean? What would change if another structure were imposed?

---

### Official Review · Reviewer_xSRc · 2023-10-31

**Soundness:** 2 fair
**Presentation:** 2 fair
**Contribution:** 1 poor
**Rating:** 1
**Confidence:** 5

**Summary:**

This paper analyzes the behavior of continuous time neural networks. The key to doing this is by analyzing the evolution of the pullback metric, which allows one to compute the intermediate metrics for visualization.

**Strengths:**

* N/A

**Weaknesses:**

* The naming of "Ricci Flow" is incorrect/misleading. Really, this is an intrinsic geometric flow, of which the Ricci Flow is a special case. This is because the intrinsic geometric flow evolves the metric according to some diff eq (in this case one parameterized by the neural network), whereas the Ricci Flow is a prescribed partial differential equation that doesn't depend on the neural network.
* The method can be simplified considerably in presentation (effectively removing most of the unnecessary manifold/homogenous space constructions). In particular, the real question is how does the jacobian $J$ evolve according to time (the other stuff is used to make sure it doesn't degenerate), as the pullback metric is just the inverse of $J^T J$ (from which one can compute the evolution $\frac{d J^T J}{dt} = J^T \frac{dJ}{dt} + \frac{dJ}{dt}^T J$), which is already well known. This is actually cleaner than the current method, which doesn't utilize the fact that $\frac{dJ}{dt}$ is known for ODEs and instead has to approximate with a step size.
* The method has an intrinsic limitation since the Jacobian scales poorly with input/output dimension.
* Experimentally, the results are only shown for extremely toy 2d data through visualization. Beyond showing ``the method can extract something", this section doesn't convey much else.

**Questions:**

N/A

---

### Meta-Review · Area_Chair_9gf2 · 2023-12-05

**Metareview:**

This paper focuses on exploring the behavior of continuous-depth neural networks, a topic that holds potential interest for the research community. However, after thorough evaluation, all reviewers have consistently found that the paper falls short of the required standards in terms of its soundness, clarity, and overall contributions. Additionally, the authors failed to participate in the rebuttal phase, missing the opportunity to address and potentially resolve the concerns raised by the reviewers. Given these significant shortcomings, I recommend rejecting this submission.

**Justification For Why Not Higher Score:**

All reviewers suggest that this paper is a clear reject. I agree with the concerns raised by the reviewers, and the authors didn't use the rebuttal phase to provide any further clarifications.

**Justification For Why Not Lower Score:**

N/A

---

### Decision · Program_Chairs · 2024-01-16

Reject